# NK and T Cell Immunological Signatures in Hospitalized Patients with COVID-19

**DOI:** 10.3390/cells10113182

**Published:** 2021-11-15

**Authors:** Laura Bergantini, Miriana d’Alessandro, Paolo Cameli, Dalila Cavallaro, Sara Gangi, Behar Cekorja, Piersante Sestini, Elena Bargagli

**Affiliations:** Respiratory Diseases Unit, Department of Medical Sciences, University Hospital of Siena (Azienda Ospedaliera Universitaria Senese. AOUS), Viale Bracci, 53100 Siena, Italy; dalessandro.miriana@gmail.com (M.d.); paolocameli88@gmail.com (P.C.); dalila.cavallaro@student.unisi.it (D.C.); sara.gangi@student.unisi.it (S.G.); beharce@hotmail.it (B.C.); sestini@unisi.it (P.S.); bargagli2@gmail.com (E.B.)

**Keywords:** NK cells, T cells, immunology, lung transplant, cytomegalovirus

## Abstract

Severe acute respiratory syndrome caused by coronavirus 2 emerged in Wuhan (China) in December 2019 and has severely challenged the human population. NK and T cells are involved in the progression of COVID-19 infection through the ability of NK cells to modulate T-cell responses, and by the stimulation of cytokine release. No detailed investigation of the NK cell landscape in clinical SARS-CoV-2 infection has yet been reported. A total of 35 COVID-19 hospitalised patients were stratified for clinical severity and 17 healthy subjects were enrolled. NK cell subsets and T cell subsets were analysed with flow cytometry. Serum cytokines were detected with a bead-based multiplex assay. Fewer CD56^dim^CD16^bright^NKG2A^+^NK cells and a parallel increase in the CD56^+^CD69^+^NK, CD56^+^PD-1^+^NK, CD56^+^NKp44^+^NK subset were reported in COVID-19 than HC. A significantly higher adaptive/memory-like NK cell frequency in patients with severe disease than in those with mild and moderate phenotypes were reported. Moreover, adaptive/memory-like NK cell frequencies were significantly higher in patients who died than in survivors. Severe COVID-19 patients showed higher serum concentrations of IL-6 than mild and control groups. Direct correlation emerged for IL-6 and adaptive/memory-like NK. All these findings provide new insights into the immune response of patients with COVID-19. In particular, they demonstrate activation of NK through overexpression of CD69 and CD25 and show that PD-1 inhibitory signalling maintains an exhausted phenotype in NK cells. These results suggest that adaptive/memory-like NK cells could be the basis of promising targeted therapy for future viral infections.

## 1. Introduction

Severe acute respiratory syndrome caused by coronavirus 2 (SARS-CoV-2) emerged in Wuhan (China) in December 2019 [1] and has severely challenged the human population [2]. Patients with COVID-19 show various clinical patterns, ranging from mild to severe forms, the latter characterised by acute respiratory distress syndrome (ARDS), multi-organ failure and high risk of death [3]. Immune dysregulation and hyperactivation caused by the infection, in particular overproduction of cytokines known as the “cytokine storm”, leading to systemic hyper-inflammatory status, were extensively studied [4,5,6,7,8]. Although several studies have investigated the role of cytokines and related cell-mediated responses, little data is yet available on the role of innate immunity and its possible contribution to host responses and disease progression.

NK cells are large granular lymphocytes, considered as the cells of the innate immune system with cytolytic activity against different targets such as tumour-derived or virus-infected cells [7]. These cells release granules that contain different acid hydrolases, such as acid phosphatase, naphthyl acetate-esterase and glucuronidase [8].

Specific aspects of NK cell biology, such as maturation, diversity and adaptive capacity, have become much clearer in the last few years [9], supporting their crucial role in first-line defence against viral infections and cancer [10,11,12,13,14]. NK cells are typically classified into cytokine-producing CD56^bright^ NK cells and cytotoxic CD56^dim^ NK cells [11]. These cells are activated when their target cells do not express MHC-I molecules. In the last year, activating receptors, including NKG2C, NKG2D and DNAM-1, and natural cytotoxicity receptors, including NKp30, NKp44 and NKp46, were characterised [15,16,17,18,19,20,21]. Other receptors, including killer Ig-like receptors (KIRs) and the CD94:NKG2A heterodimer, as well as immune checkpoints, such as PD-1, express inhibition [13]. NK cells not only have the capacity to directly target and kill infected cells but can also influence adaptive T-cell responses [18,19]. The ability of NK cells to modulate T-cell responses can be mediated through direct T–NK interactions, cytokine production, or indirectly through dendritic cells and other cell types [18].

No detailed investigation of the NK cell landscape in clinical SARS-CoV-2 infection has yet been reported. Like T and B cells, NK cells appear to be fewer in number in patients developing severe COVID-19 and to overexpress NKG2A inhibitory receptor and PD-1 on the cell surface [22,23,24,25]. The aim of the present study was to clarify the role of NK cell and T cell subsets during SARS-CoV-2 infection in hospitalised patients with COVID-19, in an endeavour to discover potential relationships within the severity of the disease.

## 2. Materials and Methods

### 2.1. Study Population

Thirty-five patients hospitalised with COVID-19, confirmed by SARS-CoV-2 RNA-positive nasopharyngeal swab and seventeen healthy sex- and age-matched controls were enrolled in the study. Demographic and clinical data and radiological and immunological features, as well as serum concentrations of inflammatory biomarkers, were entered in a pre-specified electronic database. The study population was stratified into three subgroups (mild, moderate and severe) according to COVID-19 severity. The mild subgroup required pharmacological treatment and only low-flow oxygen therapy; the moderate subgroup needed high-flow oxygen therapy and/or non-invasive mechanical ventilation; severe patients were admitted to intensive care and underwent mechanical ventilation with orotracheal intubation. A key inclusion criterion for the study population was documentation of monolateral or bilateral pneumonia by chest X-ray or high resolution computed tomography (HRCT).

Blood samples for immunological studies were collected within 24–48 h of emergency room admission and before administration of any antiviral agents, steroids and/or immunosuppressants. Serum aliquots and peripheral blood were stored at −80 °C until assay.

All patients gave their written informed consent to the study that was approved by our local ethics committee (BIOBANCA-MIU-2010. MARKERLUNG 17431).

### 2.2. Preparation and Storage of PBMCs

Cytofluorimetric analysis was performed at Siena University respiratory diseases laboratory in the period April 2020 to April 2021. Briefly, the peripheral blood samples were collected and processed within 8 h Ficoll density gradient separation was performed through Ficoll Histopaque®-1077 (Sigma-Aldrich, Burlington, MA, USA) as previously reported Live cell counts were performed in a Burker chamber. Aliquots of 3 × 10^6^ cells were stored in liquid nitrogen in a standardised manner as previously reported [26,27,28,29,30]. All experiments were performed after thawing under the same conditions.

### 2.3. Antibodies

The following mAbs were used to detect surface markers: anti-CD3 APC-Cy7 (clone: UCHT1), CD14 APC-Cy7 (HCD14), CD16 BV510 (3G8), CD19 APC-Cy7 (HIB19), CD25 BV421 (BC96), CD69 BV421 (FN50), CD56 PE-Cy7 (HCD56), CD4 FITC (RPAT4) and CD62L (DREG-56), all IgG1 isotypes from Biolegend. CD57 VioBlue (TB03. isotype IgM) CD159C PE (REA 205, isotype IgG1), CD279 PE (PD1.3.1.3. isotype: IgG2B), CD8 Vioblue (REA734 isotype IgG1) and CD45RA PE-Vio770 (REA 562. isotype IgG1) were from Miltenyi Biotech Bergisch, Gladbach, Germany, CD158a FITC (HP-3E4 isotype IgM), CD158b FITC (CH-L. isotype IgG2b) and CD27 BV510 (L128. isotype IgG1) were from BD Biosciences. CA. USA. CD159a APC (Z199. isotype IgG2b) and CD336 PE (Z231. Isotype IgG1) were from Beckman Coulter, (Beckman Coulter. 250 S. Kraemer Blvd, Brea, CA 92821, USA).

### 2.4. Flow Cytometry Analysis of NK and T Cells

For multiparametric flow cytometry analysis, we used a standard staining protocol for extracellular markers [26]. Briefly, cells were washed with wash buffer (HBSS−/− with 2% FBS) and incubated with Ab mix for 30 min in the dark at RT. Samples were acquired using BD FACS Canto II (BD Biosciences). The optimal concentration of all Abs used in the study was defined by titration. In line with the guidelines for accurate multicolour flow cytometry analysis, we used fluorescence minus one (FMO) controls.

For analysis of NK cells, three tubes were processed. All tubes contained the following mix of mAbs: CD3APC-Cy7, CD14APC-Cy7, CD16BV510, CD19APC-Cy7. CD56PE-Cy7 (HCD56), KIR2DL1 (CD158a) FITC, KIR2DL2 (CD158b) FITC and NKG2A (CD159a) APC.

CD57 VioBlue and NKG2C (CD159C) PE. CD69 BV421 and PD-1 (CD279) PE, and CD25 BV421 and NKp44 (CD336) PE were added to the first, second and third tubes. respectively. We used a panel of anti-CD3 APC-Cy7. CD4 FITC. CD62L PE. CD8 Vioblue. CD45RA PE-Vio770 and CD27 BV510 to detect T-cell maturation.

We studied maturation of CD4^+^ and CD8^+^ T cells. A first gate was set to “forward” and “side” scatter, and a second gate was set to separate CD3+ cells. Among CD3+ we identified CD4^+^ and CD8^+^ lymphocytes. From the CD4+ population, we selected CD45RA^−^ cells. T central memory (Tcm) phenotypes (CD4^+^CD62L^++^CD27^++^) and T effector cells (CD4^+^CD62L^−^CD27^+/−^). From the CD8+ population, we selected CD45RA^−^ cells. T central memory (Tcm) phenotypes (CD8^+^ CD62L^++^CD27^++^) and T effector cells (CD8^+^ CD62L^−^CD27^+/−^). As well, CD4+ T and CD8+ T cells clustered as T naive (Tn) and T stem cell memory (T_SCM_) based on expression of CD45RA: CD4^+^CD45RA^+^CD27^+++^CD62L^+++^ (CD4Tn/scm) and CD8^+^CD45RA^+^CD27^+++^CD62L^+++^ (CD8Tn/scm). All gate strategy were reported in Appendix A.

### 2.5. Immunoassay

Serum concentrations of biomarkers including IL-6, sFAS, sFASL, granzyme A, granzyme B, perforin and granulysin were quantified by bead-based multiplex LEGENDplex™ analysis (LEGENDplex™ Custom Human Assay Biolegend. 8999 BioLegend Way, San Diego, CA 92121, USA) according to the manufacturer’s instructions. Reactions were run in duplicate with a BD FACSCantoII flow cytometer (BD Biosciences, San Jose, CA, USA). The data were processed with Legendplex V8.0 software (Biolegend) and concentrations were expressed in pg/mL.

## 3. Statistical Analysis

The results were expressed as means and standard deviations (SD) or medians and quartiles (25th and 75th percentiles) for continuous variables, as appropriate. One-way ANOVA nonparametric test (Kruskal–Wallis test) and Dunn test were used for multiple comparisons, while the Mann–Whitney U test was used for two-group comparisons. The Chi-squared test was used for categorical variables. Sensitivity and specificity by ROC curve were used to describe accuracy for the different cell subsets. The Youden index (J = max (sensitivity+specificity-1)) was used to establish the best cut-offs. The Spearman test was used to look for correlations between immunological and clinical data. A *p*-value less than 0.05 was considered statistically significant. Statistical analysis and graphic representation of the data were performed by GraphPad Prism 9.0 software.

Principal Components Analysis with heatmap was performed using BioVinci software (BioTuring Inc., San Diego, CA, USA). PCA was used to reduce the dimensionality of the data hyperspace and Hierarchical Heatmaps was used for sample clustering based on cell composition. The cell subsets of patients were used to create a decision tree model to determine the best clustering variables according to the Gini criterion.

## 4. Results

### 4.1. Data of Patients

Demographic data, blood cell count, clinical data, radiological features and immunological findings are reported in Table 1. There was a prevalence of males in the groups of patients (28/35).

Regarding clinical status. 85% of patients showed at least two symptoms at onset. fever being the most common (86%). Only four patients (12.5%) did not have a history of any medical or surgical comorbidity.

Regarding blood parameters, neutrophil and lymphocyte counts were significantly lower in severe COVID-19 patients than in the other groups (*p* = 0.002 and *p* = 0.03. respectively), whereas CRP was significantly higher in severe patients than in the other severity groups (*p* = 0.012).

### 4.2. Differences in NK and T Cells in Hospitalised Patients with COVID-19 and Controls

Regarding NK cell phenotype compared with healthy controls, patients with COVID-19 showed fewer immature CD56^bright^CD16^neg^ NK cells (*p* < 0.0001) and maturing CD56^dim^CD16^bright^NKG2A^+^NK cells (*p* = 0.0437) (Figure 1a,b) and a parallel increase in the CD56^dim^CD16^bright^ NK subset (*p* = 0.0057) (Figure 1c,d).

Interestingly, there were statistically significant increments in CD56^dim^ NK cell activation marker CD69 C-type lectin (*p* = 0.038) (Figure 1e,f) and in CD56^dim^ NK cells expressing the checkpoint molecule PD-1 (*p* = 0.040) in COVID-19 patients with respect to controls (Figure 1g,h). Higher percentages of CD25 in CD56^bright^ (*p* < 0.0001) and CD56^dim^ (*p* < 0.0001) were found in patients (Figure 1i,l). The same pattern emerged for NKp44 in CD56^bright^ (*p* < 0.0001) and CD56^dim^ (*p* < 0.0001) in patients (Figure 1m).

Frequencies of CD3^+^ T cells and different clusters of T-cell differentiation were also explored. Notably, CD3^+^ and CD8^+^ T cells were significantly fewer in COVID-19 patients than controls, (*p* < 0.0001 and *p* = 0.0104 respectively). The same trend was recorded for CD4^+^CD45RA^+^, CD4^+^TCM, CD4^+^TSCM, CD8^+^CD45RA^+^ and CD8^+^TCM (*p* = 0.0036. *p* = 0.0007. *p* < 0.0001. *p* = 0.0058. *p* = 0.0014. respectively). On the contrary, CD4^+^CD45RA^−^, CD8^+^CD45RA^−^ and CD8^+^Teff were significantly more abundant (*p* = 0.0045. *p* = 0.0061 and *p* = 0.0470) in patients than controls (Figure 2).

Differently expressed immunological cell subsets were added to the PCA analysis in order to highlight populations differentiating COVID-19 patients from controls.

The first and second principal components explained 33.12% and 13.98% of the total variance. CD56^dim^CD25^+^ being the most discriminatory variable, as confirmed by decision tree analysis to determine which variables clustered best by the Gini criterion (Figure 3a,b).

### 4.3. Differences in NK and T Cells in Mild. Moderate and Severe Hospitalised Patients with COVID-19

After stratification of our cohort according to the clinical severity criteria described in Methods. we observed significantly higher adaptive/memory-like NK cell frequencies in patients with severe disease than in those with mild and moderate phenotypes (*p* = 0.0080) (Figure 4). Concerning T cell analysis CD3^+^. CD8TCM and CD8TSCM cells were significantly less frequent in severe COVID-19 patients than in the mild and moderate groups (*p* = 0.0476. *p* = 0.0236 and 0.0331), whereas CD8Teff cell percentages were significantly more frequent in severe COVID-19 patients than in the mild and moderate groups (*p* = 0.0024) (Figure 4).

Unsupervised PCA analysis was applied to discriminate the mild. moderate and severe COVID-19 populations. The first and second principal components explained 44.14% and 22.55% of the total variance, adaptive/memory-like NK cells being the most discriminatory variable. as confirmed by decision tree analysis to determine which variables clustered best by the Gini criterion (Figure 5a,b).

### 4.4. Differences in NK and T Cells in Hospitalised Patients in Relation to Death and Survival

Six of the 35 patients enrolled in the study died. CD56^dim^CD16^bright^ (*p* = 0.0301) and adaptive/memory-like NK cell frequencies were significantly higher in patients who died than in survivors (*p* = 0.0007). Regarding T cells, CD8TCM were less frequent (*p* = 0.048) and CD8Teff were more frequent (*p* = 0.0044) in patients who died than in survivors. Supervised PCA on these cell subsets showed that the first and second principal components explained 63% and 21.5% of the total variance, adaptive/memory-like NK cells being the most discriminatory variable, as confirmed by decision tree analysis to determine which variables clustered best according to the Gini criterion and by heatmap analysis (Figure 5c,d).

ROC curves were plotted to assess the discriminatory values of cell percentages and to determine sufficiently sensitive and specific cut-off values. CD56^dim^CD16^bricht^ NK cells showed the best areas under the curve (AUC) when we compared surviving and non-surviving COVID-19 patients (AUC = 0.78. 95%CI: 0.62–0.93; Cut-off: 67.7% spec.:65.5% sens.:100% *p* = 0.0320). The same pattern was recorded for memory-like NK cells (AUC = 0.91. 95%CI: 0.80–1; Cut-off: 28.79% spec.:82% sens.:100% *p* = 0.0018) and CD8Teff (AUC = 0.85. 95%CI: 0.72–0.98; Cut-off: 97.6% spec.:69% sens.:100% *p* = 0.0071).

### 4.5. Serum Inflammatory Cytokines

We examined inflammatory cytokines in the serum of all patients and controls enrolled in the study. The correlations between cytokines are reported in a correlation matrix (Figure 6a). where we observe a direct correlation between perforin and granzyme b (*p* < 0.0001 r = 0.97) and granulysin (*p* < 0.0001 r = 0.97). The only significant inverse correlation was between granzyme A and IL-6 (*p* = 0.037 r = -0.4). Severe COVID-19 patients showed higher serum concentrations of IL-6 than mild and control groups (Figure 6b). We found significantly higher serum concentrations of IL-6, sFas, granzyme B and perforin in patients than controls, and lower sFasL and granulysin (Figure 6a). Serum concentrations of sFas were higher in severe patients than controls. whereas those of sFasL and granulysin were lower in severe COVID-19 patients than controls.

Regarding correlations between serum concentrations of cytokines and cell percentages. direct correlations emerged for IL-6 and adaptive/memory-like NK (*p* = 0.02 r = 0.41). CD3^+^ (*p* = 0.02 r = 0.42) and CD8Teff (*p* = 0.039 r = 0.38). Immature NK percentages showed a direct correlation with serum concentrations of granzyme B and perforin (*p* = 0.01 r = 0.47 and *p* = 0.01 r = 0.45. respectively). Serum concentrations of sFasL showed a direct correlation with maturing NK (*p* = 0.04 r = 0.37) and CD56^+^NKp44 (*p* = 0.01 r = 0.42) cell percentages.

## 5. Discussion

The immune responses caused by SARS-CoV-2 infection are not yet fully understood, but a major alteration in lymphocyte number and activity has repeatedly been reported in patients with severe disease. The aim of the present study was to obtain further insights into the dysregulation of lymphocyte subsets, including NK and T cell maturation and progression in patients with different severities of COVID-19. Our results provide new details on NK and T cell responses in hospitalised patients with SARS-CoV-2 infection.

Our study demonstrated dysregulation of immune responses in our patients, which suggests that surveillance of lymphocyte subsets may be useful in the clinical management of COVID-19 patients. These data are also in agreement with the immunological changes recently described in a patient with mild-to-moderate COVID-19 patient that required hospitalisation and showed a significant increase in activated T cells [31,32,33]. Moreover, NK cells in COVID-19 patients resulted highly activated. This is demonstrated through the overexpression of CD25, CD69 and NKp44 on the cell surface. While for T cells, CD8 plays a crucial role, demonstrating a lower proportion in severe COVID-19 patients than in mild to moderate, although the behaviour of Teff is the opposite, suggesting that different behaviour define the different progression of diseases.

Unsupervised high-dimensional PCA cluster analysis of NK and T cells identified CD56^dim^CD25^+^ as the best variable for distinguishing COVID-19 patients and controls, as confirmed by decision tree analysis. In COVID-19 patients, high serum concentrations of CD25 contribute to lymphopenia [34] and therefore serum concentrations of soluble CD25 were also high, leading to the proliferation of pro-inflammatory T cells that aggravate disease severity [35].

Interestingly. CD56^bright^CD16^neg/dim^ NK cells are considered to be efficient cytokine producers with immunoregulatory properties [36], whereas CD56^dim^CD16^bright^ NK cells have essentially cytotoxic functions [37]. In line with previously published paper [32], we observed an increase in these cytotoxic CD56^dim^CD16^bright^ cell phenotypes, associated with a decrease in immunoregulatory subsets of CD56^bright^CD16^neg/dim^ NK cells in COVID-19 patients. This evidence of enhanced expression of cytotoxic NK subpopulations was further supported by elevated serum levels of sFas granzyme B and perforin that we found in our COVID-19 cohort. Lorente et al. reported that soluble levels of sFas, a proapoptotic protein of the extrinsic pathway. were lower in patients who survived COVID-19 [38]. Interestingly, in line with our data, Cifaldi et al. demonstrated that IL-6 directly reduced the expression of perforin and granzyme B [39]. As already demonstrated by Mazzoni et al. we also found an inverse correlation between serum levels of IL-6 and granzyme A, particularly in intensive-care patients [40].

Our results also sustain the hypothesis of a hyperactivation state of NK cells due to overexpression of CD25 and CD69 on the cell surface. The human CD69 antigen is one of the first cell-surface molecules expressed after activation of T, B and NK lymphocytes. CD69 is rapidly induced in NK cells shortly after activation and its role in NK cytotoxicity was demonstrated [41].

Higher percentages of adaptive/memory-like NK cells in patients with severe phenotypes compared to those with mild and moderate disease emerged from our data. In particular, cytotoxic CD56^dim^CD16^bright^ and adaptive/memory-like NK cells were significantly more frequent in patients who died than in survivors. Adaptive/memory-like NK cell proliferation, characterised by high expression of NKG2C CD57, was originally described in response to cytomegalovirus infection [42]. Interestingly, our results also appear to confirm a reliable increase in these subsets of NK cells in patients with SARS-CoV-2 infection, suggesting intriguing insights into the role of these cells in the modulation of the immune response against viral infection.

Soleimanian et al. suggested that adaptive/memory-like NK cells, in particular NKG2C, could be exploited in future viral immunotherapy for COVID-19 [43], while Maucourant et al. also reported that the hallmarks of different immunotypes of patients were a high expression of perforin, NKG2C and Ksp37 reflecting the increased presence of adaptive NK cells in the bloodstream of patients with severe disease [44]. However, further studies are needed to determine the specific contribution of adaptive/memory-like NK cell proliferation to COVID-19 pathogenesis [45].

We also observed enhanced expression of programmed death-1 (PD-1) in CD56^dim^ NK cells. PD-1 is an immunoregulatory receptor that may be expressed by different immune cell subtypes, including NK cells and T cells [46]. Dysregulation of PD-1 on the surface of T cells has been widely investigated [47], but little data is yet available about the expression of checkpoint inhibitors on the NK cell surface. In any case, there is evidence that PD-1 inhibitory signalling properties maintain an exhausted phenotype in NK cells and T cells in chronic infection and cancer [48]. In line with Li et al. our study found that expression levels of PD-1 on NK cells were significantly elevated in patients with COVID-19 compared to healthy controls, which implies an exhausted state of NK cells in patients with COVID-19 [37].

The natural cytotoxicity receptor NKp44 is another activating receptor that plays a crucial role in most functions exerted by activated NK cells [49]. Its role in COVID-19 has not previously been investigated. Here for the first time, we show higher percentages of this molecule in COVID-19 patients than controls, demonstrating important activation of NK cell cytotoxicity in this infection.

As Influenza Virus. SARS-CoV-2 directly affects NK cells. Peripheral blood CD56dimCD16+ and CD56brightCD16− NK cells were primed during influenza A infection [50]. As in SARS-CoV-2, Influenza-specific memory NK cells in humans suggest influenza-specific responses mediated by different subsets of NK cells [51].

Regarding T cells, a key role was performed by CD8, CD8 T cells and their various subsets, including CD8Teff and CD8Tscm, were higher in controls and in patients with mild COVID-19 than in patients with moderate or severe disease. This interesting finding is in line with Uzhachenko et al. who demonstrated that cytotoxic NK and adaptive CD8+ T lymphocytes (CTL) interact to elicit specific cytolytic outcomes in viral infection [52] and may therefore significantly influence the clinical course of COVID-19. Jiang reported that although CD8^+^ T and NK cells decreased in absolute number they increased in cytotoxic potential [53].

Regarding comorbidities, several pieces of evidence showed that immunometobolic dysregulation exaggerated by both obesity and viral infection can affect the progression of COVID-19, leading to a severe course [54,55]. Although the HC cohort was sex-matched, these subjects do not show any comorbidities or chronic diseases and this can be considered a bias of the study.

In conclusion, all these findings provide new insights into the immune response of patients with COVID-19. In particular, they demonstrate activation of NK through overexpression of CD69 and CD25 and show that PD-1 inhibitory signalling maintains an exhausted phenotype in NK cells. These results suggest that adaptive/memory-like NK cells could be the basis of promising targeted therapy for future viral infections.

## Figures and Tables

**Figure 1 cells-10-03182-f001:**
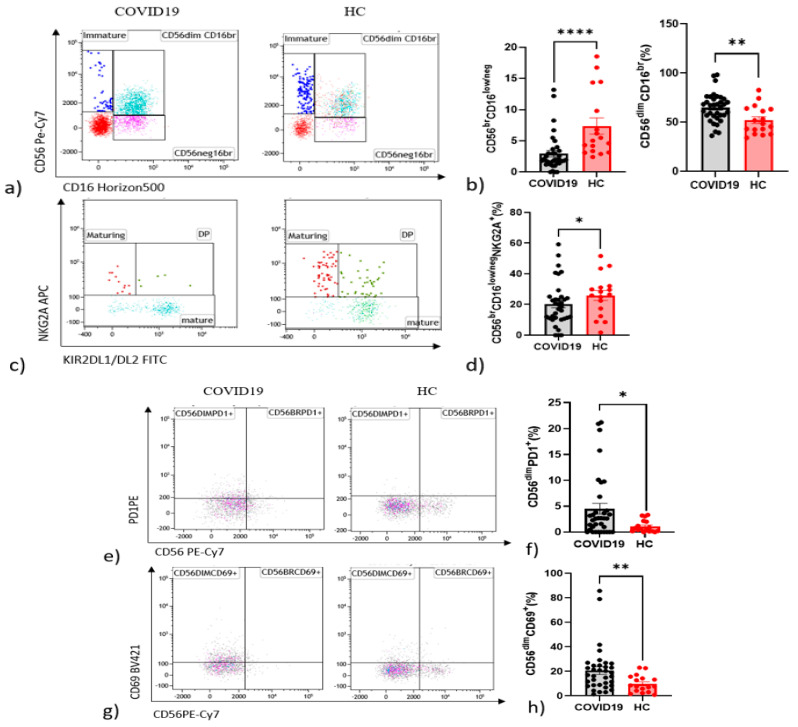
Alteration in the proportion of NK cell subsets expressing in peripheral blood NK cells from COVID-19 patients and HC. (**a**) Frequency of CD56^bright^CD16^low^ Immature NK. CD56^dim^CD16^bright^ Mature NK cell and CD56^neg^CD16^bright^ NK cells subsets among groups. (**b**) Histogram of comparisons of cell expression in NK cells among groups. (**c**) Frequency of NKG2A^+^KIR^−^ Maturing NKG2A^+^KIR^+^ DP NK cell and NKG2A^−^KIR^+^ NK subsets among groups. (**d**) Histogram of comparisons of cell expression NKG2A and KIR on NK cells among groups. (**e**) Frequency of PD1+NK cell subsets among groups. (**f**) Histogram of comparisons of cell expression in PD1+NK cells among groups. (**g**) Frequency of CD69+NK cell subsets among groups. (**h**) Histogram of comparisons of cell expression in CD69+NK cells among groups. (**i**) Frequency of NKp44^+^ NK cell subsets among groups. (**l**) Histogram of comparisons of cell expression in NKp44^+^NK cells among groups. (**m**) Frequency of CD25^+^NK cell subsets among groups. (**n**) Histogram of comparisons of cell expression in CD25^+^NK cells among groups. * *p* < 0.05. ** *p* < 0.01, **** *p* < 0.0001.

**Figure 2 cells-10-03182-f002:**
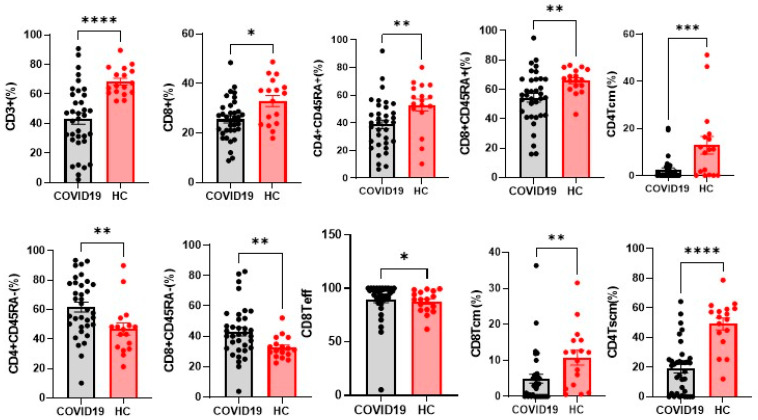
Alteration of immunologic features of the patients with COVI-19. Percentages of T cells from the healthy controls (n = 17), and COVID-19 (n = 35). The level of significance is indicated as follows: * *p* < 0.05. ** *p* < 0.01, *** *p* < 0.001, **** *p* < 0.0001.

**Figure 3 cells-10-03182-f003:**
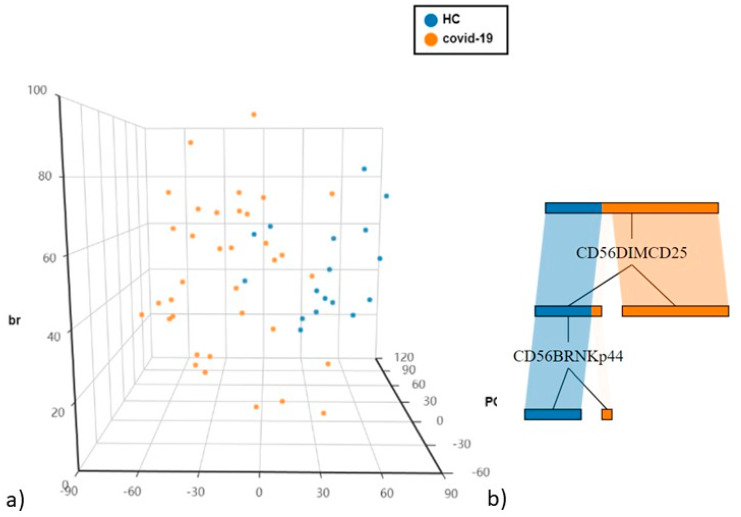
(**a**) Principal Component Analysis (PCA) of NK cells and T cell subsets (**b**) The cellular subsets of patients were employed to create a decision tree model for the detection of best clustering variables between the two groups.

**Figure 4 cells-10-03182-f004:**
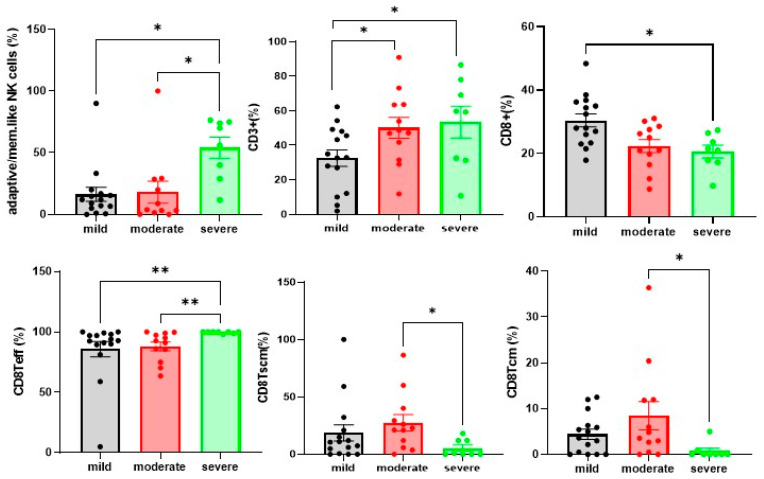
COVID-19 patients showed an imbalanced proportion of T cell subsets in peripheral blood T cells. Percentages of adaptive/mem.like NK cells. CD3^+^T cells. CD8^+^T cell subsets of total PBMCs from mild cases (n = 15). moderate cases (n = 12). and severe cases (n = 8). The level of significance is indicated as follows: * *p* < 0.05. ** *p* < 0.01.

**Figure 5 cells-10-03182-f005:**
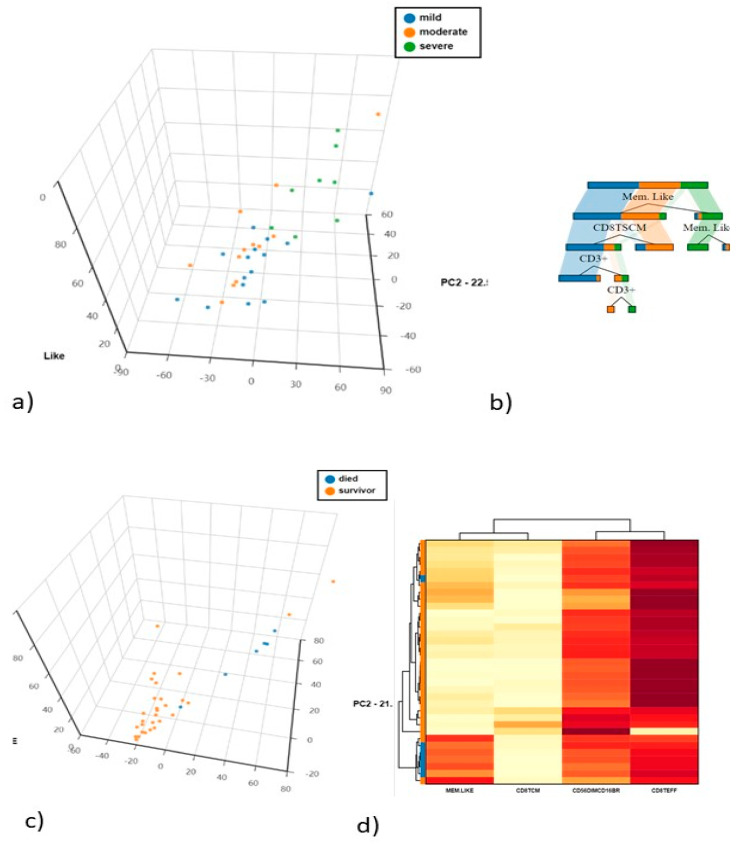
(**a**) Principal Component Analysis (PCA) of NK cells and T cell subsets (**b**) The cellular subsets of patients were employed to create a decision tree model for the detection of best clustering variables between the two groups. (**c**) Principal Component Analysis (PCA) of NK cells and T cell subsets between died and survivors COVID-19 patients (**d**) The heat map analysis of the best discriminatory variable between died and survivors COVID-19 patients.

**Figure 6 cells-10-03182-f006:**
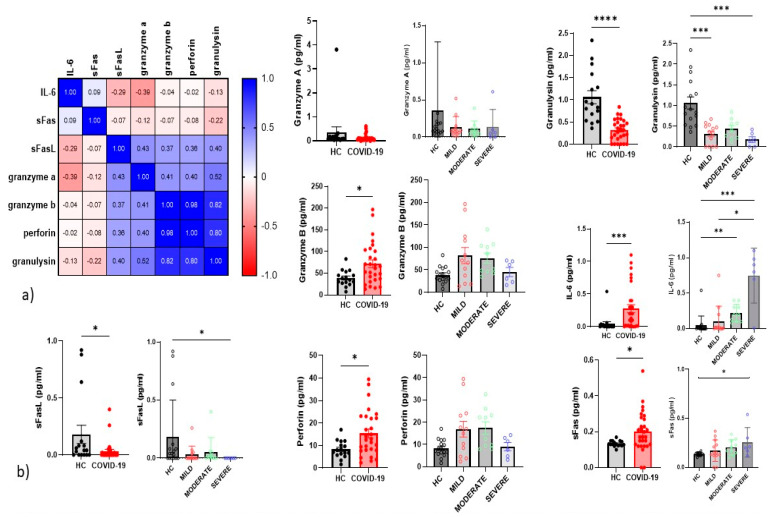
(**a**) Correlation Matrix of cytokines (**b**) Serum cytokine levels from the healthy controls (n = 17). and all COVID-19 patients (n = 35) and among mild cases (n = 15). moderate cases (n = 12) and severe cases (n = 8). The level of significance is indicated as follows: ns. notsignificant; * *p* < 0.05. ** *p* < 0.01, *** *p* < 0.001, and **** *p* < 0.0001.

**Table 1 cells-10-03182-t001:** Demographic data and blood parameters of COVID-19 cohort and healthy controls.

	HC (n = 17)	COVID-19 (n = 35)	*p* Values	Mild (n = 15)	Moderate (n = 12)	Severe (n = 8)	*p* Values
**AGE (years)**	65.3 ± 13.7	68.2 ± 16.3	ns	70.8 ± 14	58 ± 7	71.5 ± 120	ns
**GENDER (f/m)**	6/11	7/28	ns	2/13	4/8	1/7	ns
**SMOKING HABITS (current/never/former)**	6/6/5	11/9/15	ns	3/4/8	4/3/5	4/2/2	ns
**Chest X-ray** **(monolateral/bilateral pneumoniae)**		10/25		5/10	4/8	1/7	ns
**Comorbidities:** **yes/no** **diabetes** **arterial hypertension** **other lung diseases** **cancer** **haematological disorders**	0/35	31/4		12/3 6 8 5 2 1	11/1 4 9 3 4 0	8/0 4 7 4 3 1	ns
**CRP (mg/L) (median (*IQR*))**		4 (2.2–9.9)		3.8 (2.1–7.2)	3.1(2–4.4)	5.4(3.8–7.3)	0.012
**Blood counts:** **RBC** **WBC** **PLT**		4.4 (4.1–4.8) 6 (4.1–7.7) 197.5 (158–244)		4.2 (4.1–4.4) 6 (4.5–7.5) 197 (162–244)	3.9 (3–5.1) 6.7 (4.5–8.7) 217 (190–247)	3.5 (3–4.3) 5.3 (4.5–6.4) 190 (161–251)	ns ns ns
**Leucocytes counts (%):** **Lymphocytes** **Neutrophils** **Eosinophils** **Monocytes**		15.9 (9.3–19.7) 77.6 (70.9–88.5) 0 (0–0.2) 5.3 (6.1–10.1)		16.8 (9.5–28.9) 77.6 (56.5–80) 0 (0–0.4) 8.8 (4.1–12.8)	17 (10.5–32.9) 78.7 (52.1–82) 0 (0–0.4) 8.8 (4.1–12.8)	11.1 (7–14.6) 64 (48.3–83.2) 0 (0–0.6) 7.6 (5.1–11.1)	0.002 0.03 ns ns

## Data Availability

The data presented in this study are available on request from the corresponding author.

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
