# Peer review of "NK and T Cell Immunological Signatures in Hospitalized Patients with COVID-19"

_cells, 2021, doi:10.3390/cells10113182_

Round 1

Reviewer 1 Report

This is an excellent study delivering many novel information regarding the role of NK and T lymphocytes during COVID-19 infection. The phenotypic analysis of both cell types are very detailed and complex. However, some minor changes and supplementary informations could further improve the quality of the paper.

  1. Lines 44-45: Before describing NK cells in detail, please make a short overview of antiviral immune responses in general.
  2. Line 67: was COVID-19 severity determined only based on oxygen requirement of the patient? Did the patients have chest CT examinations and scoring?
  3. Line 147, Table 1: please list the comorbidities which were taken into account.
  4. Line 155: please provide representative dot blots for gating strategies for NK and T lymphocytes.
  5. The size of figures should be increased.
  6. Lines 272 and 278: a short hypothetical summery and author's opinion is needed regarding the possible role of the investigated lymphocytes during COVID-19 infection (e.g. which findings are thought to be physiological, compensatory or pathological)
  7. Are there any data about characteristics of NK and T cells in other viral pulmonary infections for a comparison (e.g. Influenza A)? 

Author Response

1. Lines 44 45: Before describing NK cells in detail, please make a short overview of antiviral immune responses in general.
Thank you for the suggestion. We provided a short overview of antiviral immune responses in general, before the detailed NK cell description as follows:
“NK cells are largr4e granular lymphocytes, considered as the cells of the innate immune system with cytolytic activity against different targets such as tumor-derived or virus-infected cells (1). These cells release granules contain different acid hydrolases, such as acid phosphatase, naphthyl acetate-esterase and glucuronidase (2).”

2. Line 67: was COVID-19 severity determined only based on oxygen requirement of the patient? Did the patients have chest CT examinations and scoring?
Thanks for your comment. The COVID19 severity was determined on the basis of severity of respiratory failure and need for mechanical ventilation and ICU hospitalization, as suggested by WHO. The chest HRCT was not performed in every patient, as well as a CT score for this radiological analysis. However, all of patients performed chest X-ray and we added these findings in the Table.

3. Line 147, Table 1: please list the comorbidities which were taken into account.
Thank you for the suggestion. We added in table 1 the main comorbidities of our COVID19 patients: hypertension and diabetes mellitus, and their prevalence in our population.

4. Line 155: please provide representative dot blots for gating strategies for NK and T lymphocytes.
Thank you for the comment. We provided a representative dot blot for NK and T lymphocytes gating strategy as you suggested.

5. The size of figures should be increased.
Thank you for the suggestion. We improved the size of figures.

6. Lines 272 and 278: a short hypothetical summery and author's opinion is needed regarding the possible role of the investigated lymphocytes during COVID-19 infection (e.g. which findings are thought to be physiological, compensatory or pathological).
Thank you for the suggestion. We agree with the reviewer, and we added our opinion about the usefulness of lymphocytes monitoring in COVID19 patients. Since it was depleted in severe patients on mechanical ventilation, our study demonstrated dysregulation of immune responses in our patients, which suggests that surveillance of lymphocyte subsets may be useful in the clinical management of COVID19 patients. These data are also in agreement with the immunological changes recently described in a patient with mild-to moderate COVID-19 patient that required hospitalization, and showed a significant increase in activated T cells (3). Moreover NK cells in COVID-19 patients resulted highly activated. This is demonstrated through the over expression of CD25, CD69 and NKp44 on cell surface. While for T cells, CD8 play a crucial role, demonstrating a lower proportion in severe COVID-19 patients than in mild to moderate, although the behaviour of Teff is the opposite, suggesting that different behaviour define different progression of diseases.

7. Are there any data about characteristics of NK and T cells in other viral pulmonary infections for a comparison (e.g. Influenza A)?
Thank you for the comment. We found different article that reported the role of NK and T cell maturation during influenza A infection. In particular we found the importance of memory NK also in influenza virus infection. We integrated our discussion as follow: “As Influenza Virus, Sars Cov-2 directly affects NK cells. Peripheral blood CD56dimCD16+ and CD56brightCD16− NK cells were primed during influenza A infection (41). As in Sars Cov-2, Influenza-specific memory NK cells in humans suggest influenza-specific responses mediated by different subsets of NK cells (42).”

Reviewer 2 Report

1) Please discuss differences in control and covid groups regarding chronisch health conditions.

If possible, add data from chronically ill (matched) non-covid patients.

Given that the controls were described as "healthy" makes the differences to the covid group diffucult.

Author Response

1) Please discuss differences in control and covid groups regarding chronisch health conditions. If possible, add data from chronically ill (matched) non-covid patients. Given that the controls were described as "healthy" makes the differences to the covid group diffucult.

Thank you for the comment. We added the comorbidities of our COVID-19 cohort (as you can see in table 1) in order to improve the description of our patients. Our cohort resulted sex and age matched. However, following your suggestion we also find the possible “bias” due to some comorbidities that the group of healthy can have. We noted that these in this group, 3 subject suffer of hypertension and 1 of diabetes. Due to the lower number of subject with comorbidities it is impossible to consider different group for our study. We would like in the future consider this suggestion and matche patients also by considering comorbidities and in particular chronic conditions.

Reviewer 3 Report

The study is overall well performed and presented, although figures need to be enlarged to make clearer for the reader.

Overall, there is a heavy bias towards Italian groups in the references.

Line 48 onwards: ‘In the last year’ – more up-to-date references for NKp30, NKp44, and NKp46 should be used, particularly the latest ones that mention the ligands, such as B7-H6 for NKp30 (Brandt CS et al, 2009), PDGF-D (Barrow et al. 2018) and Nidogen-1 Gaggero et al. 2018) for NKp44, and properdin for NKp46 (Narni-Mancinelli et al. 2017).

Author Response

Overall, there is a heavy bias towards Italian groups in the references.
Line 48 onwards: ‘In the last year’ – more up-to-date references for NKp30, NKp44, and NKp46 should be used, particularly the latest ones that mention the ligands, such as B7-H6 for NKp30 (Brandt CS et al, 2009), PDGF-D (Barrow et al. 2018) and Nidogen-1 Gaggero et al. 2018) for NKp44, and properdin for NKp46 (Narni-Mancinelli et al. 2017).

Thank you for the suggestion. We carefully searched and added the up-to-date references for NKp30, NKp44, and NKp46. Moreover, we added in the reference lists the citations that the reviewer suggested (Brandt CS et al, 2009; Barrow et al. 2018; Gaggero et al. 2018; Narni- Mancinelli et al. 2017)

Round 2

Reviewer 2 Report

"We added the comorbidities of our COVID-19 cohort (as you can see in table 1) in order to improve the description of our patients."

Comment:

Given that the healthy controls do not have any of these chronic conditions, I would discuss this bias in the discussion.

Maybe addition of a discussion of the in impact of a "non-COVID group with similar chronic conditions" would be helpful.

Author Response

we upload the response to reviewer
